# Endoscopic Papillary Abnormalities and Stone Recognition (EPSR) during Flexible Ureteroscopy: A Comprehensive Review

**DOI:** 10.3390/jcm10132888

**Published:** 2021-06-29

**Authors:** Christophe Almeras, Benjamin Pradere, Vincent Estrade, Paul Meria, on behalf of the Lithiasis Committee of the French Urological Association

**Affiliations:** 1Department of Urology, La Croix du Sud Clinic-RGDS, UroSud, 52 bis Chemin de Ribaute, Boite 301, 31130 Quint Fonsegrives, France; 2French Urological Association (AFU), La Maison de l’Urologie, 11 rue Viète, 31017 Paris, France; vincent.estrade@gmail.com (V.E.); paul.meria@aphp.fr (P.M.); CLAFUrologie@gmail.com; 3Department of Urology, Comprehensive Cancer Center, Medical University of Vienna, 1090 Vienna, Austria; benjaminpradere@gmail.com; 4Department of Urology, CHU Pellegrin, 33300 Bordeaux, France; 5Department of Urology, Saint Louis Hospital, Denis Diderot University, 75010 Paris, France

**Keywords:** papilla abnormalities, endoscopy, stone, kidney

## Abstract

Introduction: The increasing efficiency of the different lasers and the improved performance of endoscopic devices have led to smaller stone fragments that impact the accuracy of microscopic evaluation (morphological and infrared). Before the stone destruction, the urologist has the opportunity to analyze the stone and the papillary abnormalities endoscopically (endoscopic papillary recognition (EPR) and endoscopic stone recognition (ESR)). Our objective was to evaluate the value for those endoscopic descriptions. Methods: The MEDLINE and EMBASE databases were searched in February 2021 for studies on endoscopic papillary recognition and endoscopic stone recognition. Results: If the ESR provided information concerning the main crystallization process, EPR provided information concerning the origin of the lithogenesis and its severity. Despite many actual limitations, those complementary descriptions could support the preventive care of the stone formers in improving the diagnosis of the lithogenesis mechanism and in identifying high-risk stone formers. Conclusion: Until the development of an Artificial Intelligence recognition, the endourologist has to learn EPSR to minimize the distortion effect of the new lasers on the stone analysis and to improve care efficiency of the stone formers patients.

## 1. Introduction

The number of endoscopic treatments of urinary stones increases all over the world.

As previously demonstrated by Daudon et al. [1,2,3,4,5], the morpho-constitutional stone analysis plays a major role in identifying its etiology and thus consider its risk of recurrence. The increasing efficiency of lasers in “dusting” and “popcorning” modes [6,7,8,9] and the improved performance of endoscopic devices led to smaller stone fragments, which reduce the accuracy of the microscopic study (morphological and infrared) by the lack of components representativeness (48.6% of the stones have a mixed composition [10])). Moreover, Keller et al. [8,9] have demonstrated that laser-based Thulium fiber changed in stone composition in the infrared spectra that resulted in insufficient information of stone powder examination (Figure 1).

Since Randall’s works [11] in the 1930s, it is known that papillary calculi resulted from subepithelial lesions [12,13,14,15,16,17]. The advent quality of images with flexible retrograde ureteroscopy has allowed the in vivo description of papillary abnormalities [18,19,20,21] that can be related to various lithogenesis mechanisms [22,23,24,25,26,27,28].

Before the destruction of the stone, the urologist has the opportunity to hold a key role in stone prevention by recognizing the papillary abnormalities (endoscopic papillary recognition (EPR)) and the stone’s type (endoscopic stone recognition (ESR)).

The aim of this review was to report the current literature on endoscopic stone and papillary descriptions in order to help the urologist to improve the management of stone disease.

## 2. Methods

### 2.1. Search Strategy

The systematic review was conducted according to the preferred reporting items for systematic reviews and meta-analyses (PRISMA) extension statement. The PubMed, Cochrane library, and Embase databases were searched to identify reports published until February 2021 on endoscopic recognition of papillary abnormalities and stones by retrograde flexible ureteroscopy.

The following search terms were used: “endoscopy”, “stone”, “kidney”, and “papilla”. Manual searches of reference lists of relevant articles were also performed to identify additional studies. The primary outcome of interest was to assess the value of endoscopic description of papillary abnormalities and kidney stones in improving the diagnosis and the preventive care of stone formers.

Two investigators performed the initial screening based on the titles and abstracts of the articles to identify eligible and ineligible reports. Reasons for exclusion were noted. Potentially relevant reports were subjected to a full-text review, and the relevance of the reports was confirmed after the data extraction process. Disagreements were resolved via consensus with the co-authors and consultation of the senior author.

### 2.2. Inclusion and Exclusion Criteria

Studies were included if they included patients with urinary stones (participants) who had undergone flexible ureteroscopy with ESR or EPR (intervention) or another endoscopy procedure (comparison) to assess the effect of therapy on OS and AEs (outcome).

We excluded letters, editorials, meeting abstracts, replies from authors, case reports, and articles not published in English. References of all papers included were scanned for additional studies of interest. There was no time limitation for included studies.

### 2.3. Data Extraction

Two investigators independently extracted the following information from the included articles: first author’s name, publication year, the period of patient recruitment, number of patients, type of treatment, study design, and study funding and/or support. All discrepancies regarding data extraction were resolved by consensus with the co-authors or by discussion with the senior author.

## 3. Results

### 3.1. Study Selection and Characteristics

After a bibliographic search and the removal of duplicates, a total of 54 articles were screened.

After full text assessment, a total of 17 publications met the inclusion criteria (Figure 2 and Table 1).

### 3.2. Evidence Synthesis

Obviously, good vision represents an important condition to perform EPR-ESR. Although main papillary abnormalities could be observed with fiberoptic devices, poor image quality [34] may alter the ability of accurate diagnosis such as “intratubular plugging”. For that reason, the use of digital flexible ureteroscopes has been found to be of utmost importance to provide a better diagnosis. The percutaneous approach was generally inaccurate for EPR because of the incomplete exploration of the calices with a rigid nephroscope and its impaired vision due to bleeding and local inflammatory conditions.

### 3.3. Endoscopic Papillary Recognition (EPR)

Through the 15 studies and 1 review that met the inclusion criteria, and according to the three pathways for kidney stone formation (overgrowth on interstitial plaque, crystal deposits in renal tubules, and free solution crystallization) described by Coe et al. [14], the observation of the papillary abnormalities has the aim to determine the origin of the lithogenesis and to evaluate its severity and risk of recurrence. Because of the recent concepts of EPR and classification/grading system, no recurrence rate data according to the different endoscopic papillary abnormalities has been found in the literature at this time. Although the relation between recurrence rate and observed endoscopic abnormalities has not been clearly demonstrated yet, Strohmaier et al. [29] showed that the extent of Randall plaques (RP) was correlated with the number of calcium oxalate stone episodes (*p* = 0.012). Moreover, Ciudin et al. [35] showed that the number of papillae tip attenuation >43 HU on unenhanced abdominal CT images were correlated with stone recurrence.

EPR should be the first step during flexible ureteroscopy in view of assessing all the calyces and the papillae before impairing the vision by blood or stone dust and to avoid misleading traumatic thermal laser induced lesions that may be caused mainly by direct shot of the laser beam on the papillae. The appearance of one papilla is not accurate enough to predict the type and the severity of the disease, and a papillary abnormality could be also explained by the appearance of neighboring papillae. Almeras et al. [25] reported a mean duration of 81.4 s (range: 48–149; median 64) in exploring the entire kidney. A scoring system [20] and a classification [21] have been proposed to standardize the descriptions and store the data. As previously described and such as their quantification, the multifactorial aspect of the abnormalities needs to be described: the presence or not of calculi (anchored or intraductal) and their types, the description of the papillary lesions (that may be a cause or a consequence of the lithogenesis or stone growth), and the description of the presence of deposits (Randall’s plaques, intratubular deposits, etc.).

The prevalence of RP described during ureteroscopy in stone formers is high, ranging from 83% to 91% [19,25]. Their prevalence was impacted by the lithogenesis type: decreased in case of struvite and less present in case of intraductal crystallization [22,25,26] and CP stones [25,27]. Papilla’s percent surface area occupied by RP in stone formers differed significantly from that in non-stone formers (*p* < 0.0001) and was correlated with the number of stones [30]. Wang et al. [31] demonstrated that low-plaque Calcium Oxalate stone formers tended to be obese (50% vs. 10%; *p* = 0.03) and had a history of urinary tract infection (34% vs. 0%; *p* = 0.04).

The observation of “erosions” or “pits” at the tip of the papilla and the presence of anchored COM stones were also frequent (55.7% and 18.2%, respectively) and correlated with the amount of Randall’s plaques (Figure 3) [23,24,25]. They resulted from a dietary cause (especially low fluid intake) in most cases [36,37,38,39].

The description of intrapapillary or intraductal crystallization (Bellini plug origin) (Figure 4) was less common (15.9%) but was correlated with calcium phosphate stones (especially IVa2) and with a higher incidence of hypocitraturia (55.6%) and hypercalciuria (33%) [25]. Intraductal crystallization was related with different etiologies such as distal tubular acidosis [5] with the threat of impaired kidney function secondary to interstitial fibrosis that surrounds the Bellini ducts [17].

Nonetheless, some difficulties were reported to determine if some stones were “Randall’s Plaque-anchored” or “plug-anchored”. It has been suggested to remove the anchored stone with a basket to examine the papilla and note the presence of plugs beneath (Figure 5) [25].

The microscopic analysis of an anchored stone should also complete EPR in becoming the best method to provide a reliable analysis of the entire stone and a chance to examine the nucleus that represents one of the primary steps of kidney stone formation.

Assigned to its ability to identify the origin of crystallization (intratubular or on PR) [5,13,14,17,21,25] and the amount of the abnormalities that may predict the risk of recurrence [20,21,25], EPR is a way to understand the origin of lithogenesis to elucidate its mechanism and to improve high risk etiology diagnosis.

### 3.4. Endoscopic Stone Recognition

Only one study met the inclusion criteria for ESR. Today, 48.6% of the stones are reported to have a mixed composition (Figure 6). If the outer layers represent the most recent crystallization, they may differ from the inner part of the stone and the nucleus (Figure 7). Laser fragmentation or dusting make the “history of the stone growth” vanish and modify the proportions of fragments microscopically analyzed [8,9]. Consequently, a loss of information may arise such as a decrease in diagnosis capacities. Therefore, ESR is proposed as a useful tool to prevent the loss of information due to stone destruction.

ESR should describe the external layer (surface) of all the stones during the same procedure, including the small papillary anchored stones that represent the first steps of crystallization. The final report should also mention the polymorphism of kidney stones in case of various aspects described during the same procedure in order not to ignore a potential lithogenesis mechanism. The internal part and center of the main stone have then to be examined. To optimize the inner description, Estrade et al. [10] recommended a stone transection in two parts using Holmium laser with the following settings: frequency, 5 Hz; energy, 1.2–1.4 J; power, 6–7 W; short pulse length; fiber diameter, 230 or 270 μm. For an experienced endourologist, the concordance between the endoscopic description and the microscopic analysis was 86.1% (COM), 85% (COD), 91% (UA), 79% (CP), 65% (Brushite), 75% (Struvite), and 100% (Cystine).

The endourologist becomes the only witness able to recognize the entire stone’s aspect and the papillary abnormalities owing to his skills and his endoscopic devices. Using and reporting ESR, he also obtains the opportunity to rectify the results of the microscopic analysis according to his endoscopic descriptions.

## 4. Discussion

In the last century, eating habits have changed with an increased intake of salt, animal proteins, and refined sugar and a decreased intake of vegetables [39]. That consequently implied a change in stone composition and a prevalence increased [40]. These changes are especially concerning COM (subtype Ia [1]) that are mostly correlated with the prevalence of RP [19,25], low fluid intake [37], and the evolution of dietary habits [40,41,42].

As previously demonstrated by Daudon et al. [1,2,3,4,5], the morpho-constitutional stone analysis plays a major role in identifying the etiology of the stone disease and thus in stone recurrence. The increasing efficiency of lasers in “dusting” and “popcorning” modes [6,7,8,9] decreases the size of stone fragments and the accuracy of the microscopic study (morphological and infrared), thus impairing the etiologic investigation’s results. This lack of data may be balanced by EPR-ESR [10,18,20,25] and the papillary anchored stones analysis.

However, some limitations are still debated. First, the literature addressing the endoscopic papilla and stone recognition is poor and most of the published studies were from a single institution and had a small cohort.

As the endoscopic interpretations of the papillary abnormalities are only based on endourologist descriptions, their reliability, especially concerning the type of deposits (RP, plugs) and the origin of the crystallization (RP anchored, intraductal origin), remain a potential limitation and a potential interpretation bias [10,18,19,20,21,25,28].

The main problem in recognizing papillary abnormalities and stones composition is the very large array of descriptions and entities [20,21,25]. Thus, the learning curve for EPR and ESR is long and difficult, it has been shown that a perfect recognition of the stone was obtained in only 40.7% of the cases for urologist in training who benefited from nine specific teaching classes [43]. Nevertheless, the concordance between expert endoscopic description and microscopic analysis was much better with 86.1% (COM), 85% (COD), 91% (UA), 79% (CP), 65% (Brushite), 75% (Struvite), and 100% (Cystine) [10]. Although learning this specific skill might be time-consuming, training is certainly the key until the development of recognition models created by artificial intelligence (AI). In vitro, automatic detection of kidney stones composition from digital stone pictures has been described with a prediction of 94% (UA), 90% (COM), 86% (Struvite), 75% (Cystine), and 71% (Brushite) [44]. AI is about to be applied to in vivo validated endoscopic pictures, but stone morphological laser changes and heterogenous vision quality may hamper its development. AI will also be used to simplify EPR. Indeed, the efficacy of deep learning to segment the renal papilla, plaque, and plugs has already been described 46].

The backbone of ESR and EPR remains the recognition, which is based on a good intraoperative vision. Therefore, some variables have to be considered, such as fiberoptic devices that do not have high-definition vision quality [34], single-use and reusable digital ureteroscopes that do not seem to be equivalent in term of color, brightness, and definition [45,46,47]; and PCNL that cannot allow a complete exploration of the papillae. Today and for those reasons, the best way to proceed EPR and ESR is the use of digital flexible ureteroscopes.

Recently, it has been shown that lasers impacted the infrared analysis regarding stone composition [8,9]. Moreover, recognition could be biased by the dusting settings (high frequency and long pulse) that might change the surface appearance (Figure 8) especially due to a carbonization effect (Figure 9) (mainly described with Thulium Fiber Laser).

To limit these biases, an initial transection of the stone has been proposed but remains difficult, time consuming, and provides more fragments to treat. Therefore, the use of the fragmentation setting might help to properly assess the internal layers and the use of dusting should be used only after the complete description [8,9].

Although it represents the origin of crystallization, stone analysis and ESR often miss the nucleus structure analysis due to stone destruction. Hence, the additional EPR analysis could provide essential information regarding the lithogenesis mechanism and avoid misdiagnosis of high-risk diseases like distal tubular acidosis. Although it is still under evaluation, the intensity and the amount of the papillary abnormalities may also have a prognostic value regarding stone recurrence.

Combining these complementary methods should be gathered in a single process of endoscopic papilla and stone recognition (EPSR). It could support the preventive care of the stone formers in improving the diagnosis of the lithogenesis mechanism and in identifying the high-risk stone formers.

In this way, the urologist should play a key role in lithiasis prevention and stone formers’ care improvement.

## 5. Conclusions

The morpho-constitutional stone analysis plays a major role in identifying the etiology of the stone disease. The increasing efficiency of the lasers decreases the fragments’ size and their representativeness, induces laser-based changes in composition, and thus decreases the accuracy of the microscopic study.

The urologist has the opportunity to play a key role in stone prevention by recognizing the papillary abnormalities (endoscopic papillary recognition (EPR)) and the stone type before dusting or fragmentation (endoscopic stone recognition (ESR)). EPR and ESR should be gathered in a single process of endoscopic papilla and stone recognition (EPSR). Until the development of an AI recognition, the endourologist has to learn EPSR to minimize the distortion effect of the new lasers on the stone analysis and to improve care efficiency of the stone formers patients.

## Figures and Tables

**Figure 1 jcm-10-02888-f001:**
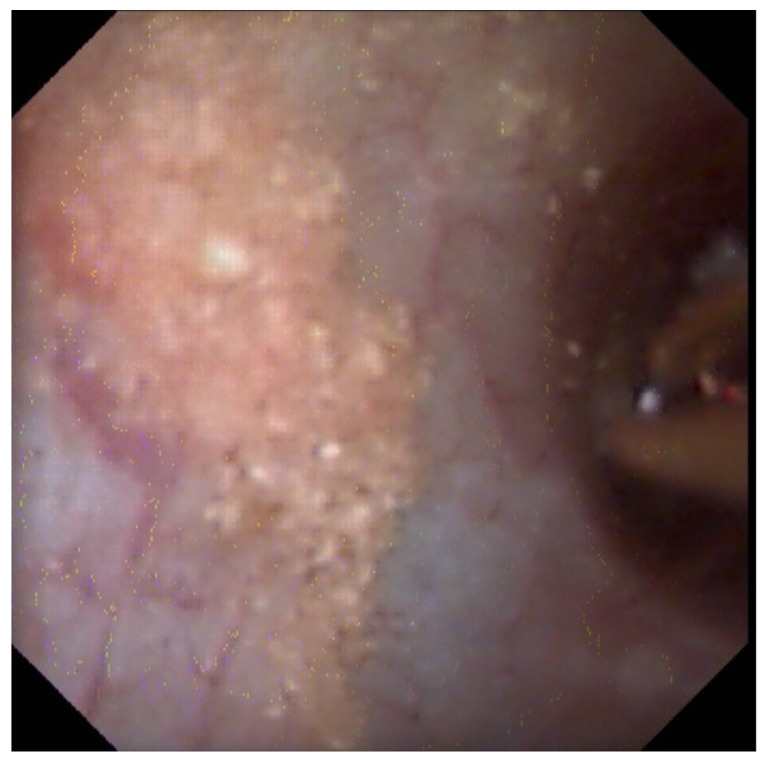
Residual “dust” after laser treatment using dusting parameters.

**Figure 2 jcm-10-02888-f002:**
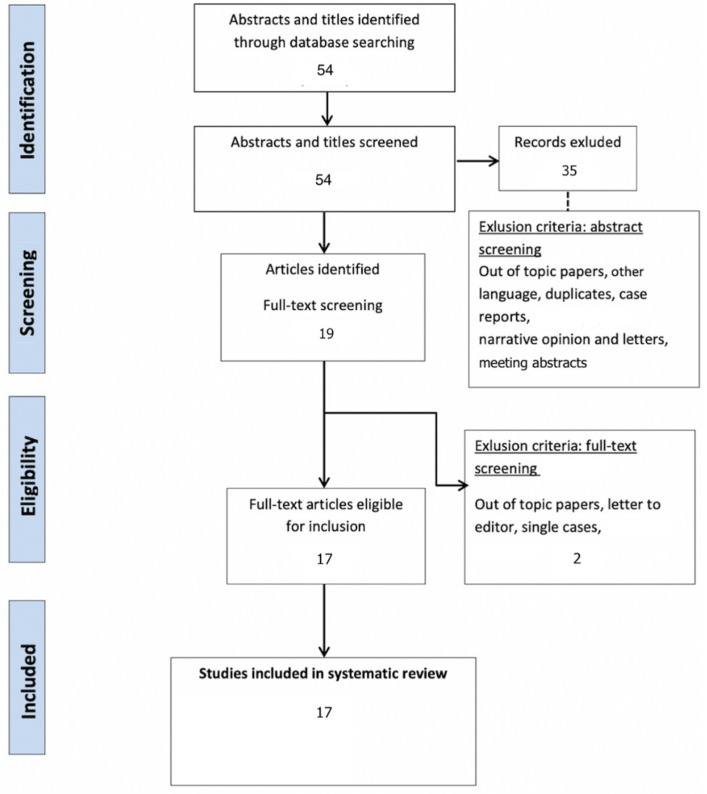
Systematic review PRISMA flow diagram.

**Figure 3 jcm-10-02888-f003:**
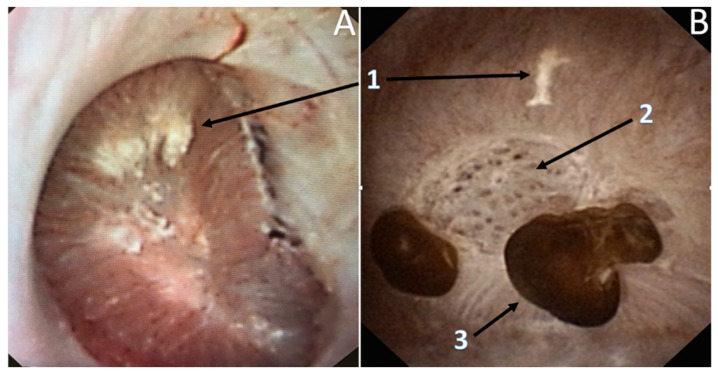
Lithogenesis on Randall’s Plaques (RP). (**A**) Papilla with RP. (**B**) Papilla with anchored stones and erosion secondary to RP. 1—RP originate from the basement membranes of thin loops of Henle, spread with CA in the surrounding interstitium, and may erode the epithelium. Their aspect is a not well delineated infiltrate of the papilla. 2—Erosion or pit, that is the footprint of a previous anchored stone drop off. 3—Typical COM anchored stones, owing to the small size of the COM crystals that are the first able to combine with the plaques.

**Figure 4 jcm-10-02888-f004:**
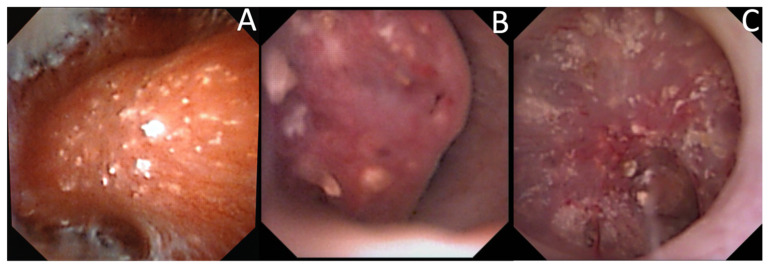
Endoscopic aspects of intratubular crystallization. Their aspect is based on very thin, small, and well-delineated deposits. (**A**) Intraductal Bellini plugs, located in the central part of the papilla. (**B**) Intraductal crystallization, with the presence of small stones in the Bellini ducts. (**C**) Intraductal crystallization and intense peripheric intraductal plugging (that may begin in the loop of Henle), with the development of nephrocalcinosis.

**Figure 5 jcm-10-02888-f005:**
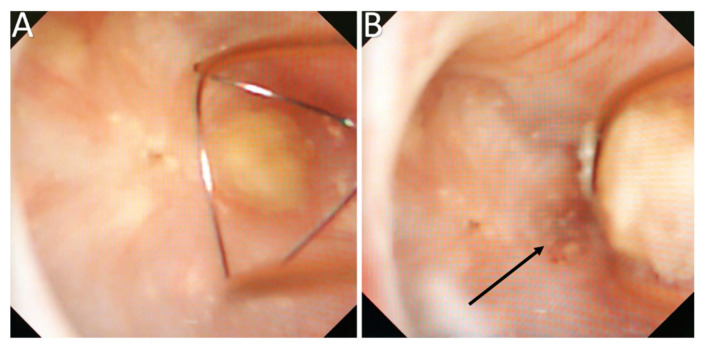
Bed observation of a removed anchored stone may reveal the presence of tubular plugs. (**A**) Observation before anchored stone retrieval. (**B**) Bed observation after anchored stone retrieval (shown by the black arrow).

**Figure 6 jcm-10-02888-f006:**
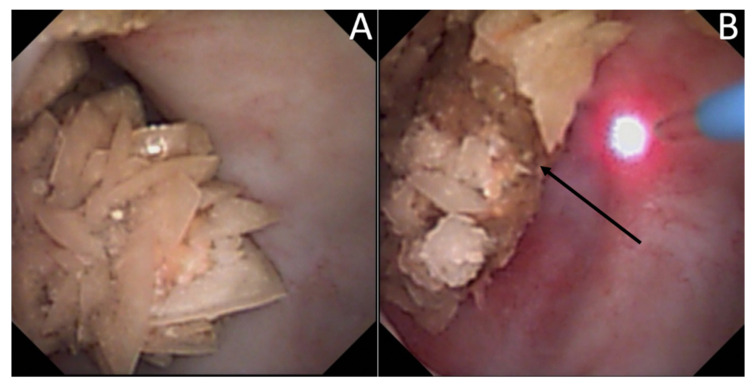
Example of a stone with a pure outer COD aspect (**A**) and during Laser treatment (**B**) the emergence of a central COM part (shown by the black arrow). After treatment, and notably, dusting of the outer part, the remaining fragments (presumably COM) will be extracted for analysis and may underestimate the whole composition of the stone in case ESR was not performed.

**Figure 7 jcm-10-02888-f007:**
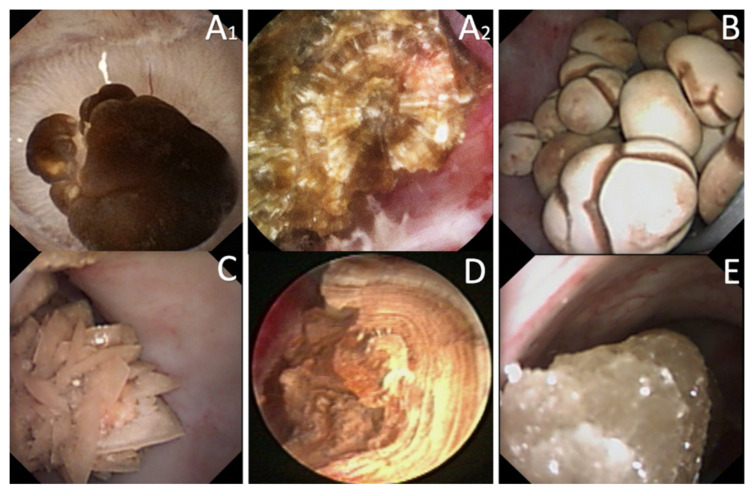
Examples of ESR. (**A1**) Typical COM stone with a smooth or mammillary dark-brown surface. In this case, papillary anchored with a RP. (**A2**) Typical transection aspect of a pure COM stone with a radiating organization of layers starting from a nucleus. (**B**) Typical COM (subtype Id [1]) stone with a pale brown-yellowish budding surface. (**C**) Typical COD stone with a yellow light spiculated surface (aggregated crystals with sharp angles and edges). (**D**) Typical UA (subtype IIIa [1]) stone with a homogeneous smooth orange surface, and after transection a concentric organization of the layers around a well-defined nucleus. (**E**) Typical Cystine stone with a bumpy or rough light brown yellow surface with a waxy aspect.

**Figure 8 jcm-10-02888-f008:**
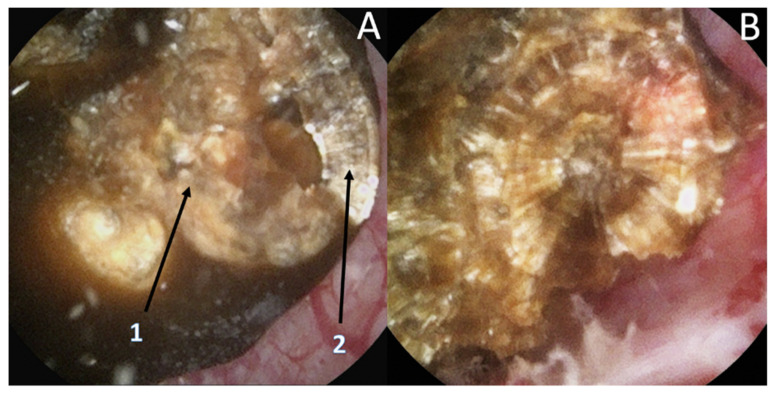
Stone morphological laser-induced changes. Example of a pure COM stone. (**A1**) After dusting settings use: Loss of the typical radiating organization of the internal layers aspect. (**A2**) After fragmentation settings use: visualization of the typical radiating organization of the internal layers. (**B**) Pure fragmentation in progress aspect: respect of the internal structure.

**Figure 9 jcm-10-02888-f009:**
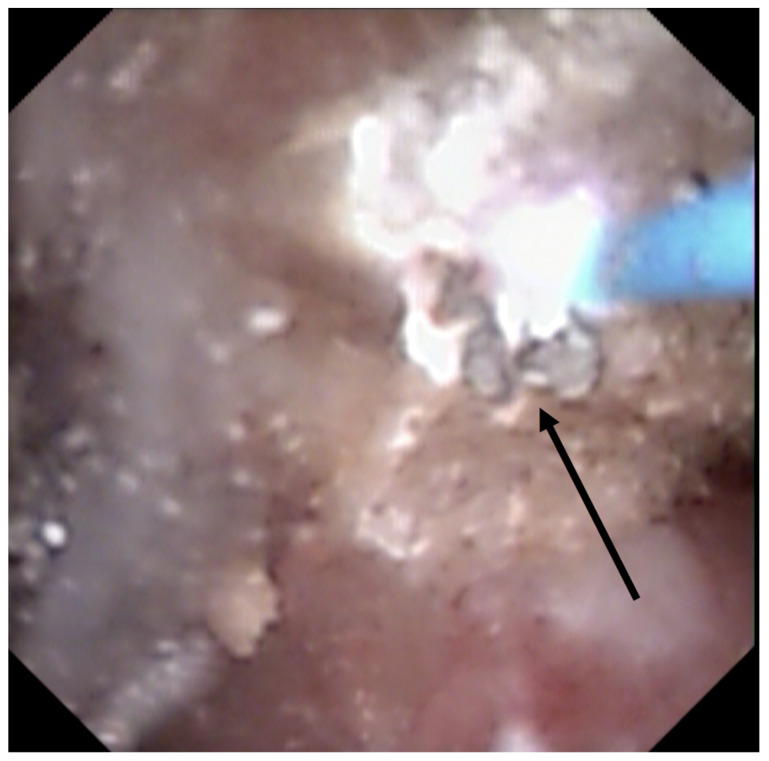
Stone morphological laser-induced changes. Carbonization (shown by the black arrow) during TFL treatment, which can be misleading for ESR.

**Table 1 jcm-10-02888-t001:** Identified and selected publications on ESR and EPR.

	Type	Subject	Number	Year
Low [18]	EPR	RP	57	1997
Darves-Bornoz [27]	EPR	RP in pediatric stone formers	8	2019
Strohmaier [29]	EPR	RP and number of stone episodes	100	2013
Kim [30]	EPR	RP and number of stones	17	2005
Wang [31]	EPR	Low RP and CaO_x_ stone formers	42	2014
Matlaga [19]	EPR	Anchored stone and RP	23	2006
Borofsky [20]	EPR	Grading Score	342	2016
Almeras [21]	EPR	Classification	164	2016
Jaeger [22]	EPR	Struvite	119	2016
Cohen [23]	EPR	Score use and correlation RP/pitting	76	2019
Borofsky [24]	EPR	Anchored stone/pitting	28	2019
Almeras [25]	EPR	Classification use and correlations RP, stones, …	88	2021
Pless [26]	EPR	Score use	46	2019
Sabaté [28]	EPR	Description	41	2020
Fernandez [32]	EPR	CP plugs detection by AI	200	2019
Estrade [10]	ESR	Correlation endoscopy/microscopy	399	2020
Marien [33]	EPR	Review	13	2016

ESR: endoscopic stone recognition; EPR: endoscopic papillary recognition; AI: artificial intelligence.

## Data Availability

Data sharing not applicable.

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
