# Peer review of "Endoscopic Papillary Abnormalities and Stone Recognition (EPSR) during Flexible Ureteroscopy: A Comprehensive Review"

_jcm, 2021, doi:10.3390/jcm10132888_

Round 1
Reviewer 1 Report
- General comments
This article is a review article about endoscopic papillary recognition and stone recognition in current era. It is nice contents. As endoscopic devices like digital scope has been advancing, we can get good view in renal collecting system, Therefore, we can see and find the details of renal papilla such as Randall plaque. In stone management, you know that the analysis of stone composition is quite important to prevent the stone recurrence post-operatively. However, stone fragments have become much tiny due to advent of novel laser system like Moses technology and Thulium laser. So, the opportunity to remove it as large stone fragment is decreasing. We, who deal with endoscopic stone management, should mind this issue.
- Comments for revisions
- Can you mention about recurrence rate according to endoscopic finding aspects? If you have this information, please describe about this. And also, mention about clinical differences as well.
- In current laser era, the thermal injury is crucial issue. Does this thermal issue influence the EPSR?? What do you think about this issue? Please mention your opinion.
Author Response
Dear Reviewer,
Thank you for your assessment and your helpful comments and suggestions.
- Can you mention about recurrence rate according to endoscopic finding aspects? If you have this information, please describe about this. And also, mention about clinical differences as well.
“Because of the recent concepts of EPR and classification/grading system, no recurrence rate data according to the different endoscopic papillary abnormalities has been found in the literature at this time.” It has been mentioned in the EPR paragraph.
- In current laser era, the thermal injury is crucial issue. Does this thermal issue influence the EPSR?? What do you think about this issue? Please mention your opinion.
“EPR should be the first step during flexible ureteroscopy in view of assessing all the calyces and the papillae before impairing the vision by blood or stone dust, and to avoid misleading traumatic thermal Laser induced lesions that may be caused mainly by direct shot of the laser beam on the papillae.” has been modified in the EPR paragraph.
Reviewer 2 Report
The review is well written and the topic is more interesting, the future prospective of this topic are analyzed by the Authors, especially the aspects linked to the limitation of stone analyses after laser stone fragmentation and the development of an artificial intelligence for recognize the EPR and ESR; and the complementary use of these informations to improve the diagnosis of the lithogenesis mechanism and in identifying the high-risk stone formers. The Authors perforemd a good literature search. The unique aspect, that the Authors could underline, is the use of the digital ureteroscope during the procedures for the quality of the images.
Author Response
Dear Reviewer,
Thank you for your assessment and your helpful comments and suggestions.
- The unique aspect, that the Authors could underline, is the use of the digital ureteroscope during the procedures for the quality of the images.
“Nowadays and for those reasons, the best way to proceed EPR and ESR is the use of digital flexible ureteroscopes.” Has been added in the discussion in the paragraph about the importance of the good vision.